# Is the Xiaomi Mi Band 4 an Accuracy Tool for Measuring Health-Related Parameters in Adults and Older People? An Original Validation Study

**DOI:** 10.3390/ijerph19031593

**Published:** 2022-01-30

**Authors:** Ana de la Casa Pérez, Pedro Ángel Latorre Román, Marcos Muñoz Jiménez, Manuel Lucena Zurita, José Alberto Laredo Aguilera, Juan Antonio Párraga Montilla, José Carlos Cabrera Linares

**Affiliations:** 1Department of Didactic of Music, Plastic and Corporal Expression, University of Jaén, 23071 Jaen, Spain; anadelacasaperez@gmail.com (A.d.l.C.P.); platorre@ujaen.es (P.Á.L.R.); jccabrer@ujaen.es (J.C.C.L.); 2Research Group HUM-790, University of Jaén, 23071 Jaen, Spain; mmjimene@ujaen.es; 3Department of Didactic of Music, Plastic and Corporal Expression, University of Sagrada Familia de Úbeda, Úbeda, 23400 Jaen, Spain; mlucena@fundacionsafa.es; 4Faculty of Physiotherapy and Nursing of Toledo, University of Castilla-La Mancha, 45005 Toledo, Spain; josealberto.laredo@uclm.es; 5Multidisciplinary Research Group in Care (IMCU), University of Castilla-La Mancha, 45005 Toledo, Spain

**Keywords:** Xiaomi Mi Band 4, laboratory and free-living condition, validity and reliability, physical activity, low-cost fitness trackers

## Abstract

Background: The aim of this study was to analyse the validity and accuracy of the low-cost Xiaomi Mi Band 4 (MB4) fitness tracker in relation to step count and heart rate in free-living conditions. Methods: 46 participants join in this study (38.65 ± 14.36 years old). The MB4 was compared with a video recording in laboratory conditions, also with the Sense Wear and Firstbeat monitors in free-living conditions. Results: No significant differences were found in the number of steps in the laboratory protocol between devices, in both, at low and high walking speed. For the free-living conditions, the MB4 showed high values of convergent validity in relation to the Firstbeat monitor during recording in both rest and walking situations. Moreover, the MB4 showed adequate values of convergent validity with the Sense Wear monitor during the 24 h recording, at medium speeds, and climbing stairs. Conclusion: The accuracy and precision of the MB4 is reasonable and can be used to monitor the average of step count and heart rate in free-living conditions.

## 1. Introduction

Physical inactivity is one of the major health problems in developed societies, and the fourth leading risk factor for global mortality [1]. To increase physical activity (PA) levels and improve health status, walking is one of the most recommended methods, since it can be performed by people of any age, as well as in any place [2]. To quantify daily PA, self-reporting has been a useful method for registering it. However, it can be inaccurate since the answer depends on the memory of the participants [3]. On the contrary, assessing PA using a valid device such as a pedometer is associated with a lower risk of all-cause mortality, with step count (SC) considered an effective way to achieve PA recommendations in adults since it is a simple, economical, and easy method for reaching these recommendations [4]. Furthermore, technological advances enable the quantifying of PA levels at a reduced cost due to the development of wearables and health applications, which have increased in both variety and quality [5,6].

Although there are no guidelines about the recommendation of the number of steps per day, Saint-Maurice et al. [7] concluded that the total number of steps per day can be more important for promoting health than step intensity. Moreover, a recent systematic review concluded that an increment of 1000 steps per day can help to reduce all-cause risk of mortality, cardio-vascular disease morbidity, and mortality, since there is an existing inverse correlation between the risk of death and PA levels (i.e., as the number of steps increases, the risk of death decreases) [8]. In this sense, a walking programme based on the number of steps assessed by a pedometer can motivate to increase PA levels in a long term, it can also provide health benefits for adults and older people [2]. Consequently, it is necessary to know the validity and reliability of these devices, since the utilization of wearables has increased for both personal and scientific purposes [9].

Recently, wearable technology has started a revolution in the Sports Technology Market since these devices are versatile and profitable due to their low prices and variety of functions [10]. Furthermore, the development and user acceptance of low-cost fitness trackers have allowed the increment of these devices to reach almost 200 million units in 2022 [11]. In consequence, low-cost PA fitness trackers have achieved great popularity in the last decade since they are considered attractive and useful tools for increasing PA levels through users’ instant feedback about their PA level and lifestyle. Additionally, they offer the possibility of sharing PA activities with people around us, which is an important motivational component in achieving PA recommendations [12,13]. In addition, fitness trackers are associated with modest changes in variables such as weight, blood pressure, and high-density lipoprotein (HDL) cholesterol [14]. These changes in weight, blood pressure, and HDL occur especially in sedentary populations who do not reach the minimum PA guidelines, since they have an extra motivation to increase their PA levels when they use a fitness tracker [15].

Due to the great variety of wearables, various brand has developed its own application, allowing users to understand the compressible data visualization that the fitness tracker has registered during a specific sport activity or daily activity. Brands as Apple, Xiaomi and Fitbit have commercialized smartwatches and fitness trackers which can synchronize with Apple Health, Mi Fit and Fitbit, respectively [16]. Nowadays, the majority of these devices (e.g., Apple Watch, Fitbit Charge, or Mi Band) are able to register physical variables, such as SC or activity intensity, sleep hours, and calories expenditure. In addition, physiological variables, such as heart rate (HR) and heart rate variability (HRV) [17]. In spite of the popularity that fitness trackers have reached in the last decade, there is not enough research to validate these devices [16]. Although some independent validation protocols have been tested in laboratory [18] and free-living conditions [19,20], there is not a standard protocol that allows for validating these devices [17].

One of the brands that have increased their sales volume in the last decade is Xiaomi. Nowadays, Xiaomi is among the top five brands, being the Mi Band series (MB) one of the most famous low-cost fitness trackers. The Mi Band 4 (MB4) is able to record daily activity for approximately 21 days. Among other functions, the MB4 registers SC, and HR. It is therefore possible to use the MB4 to monitor exercise intensity, and daily PA. Since low-cost fitness trackers are accessible to a major sector of the population and provide an incentive for increasing PA levels and improving a healthy lifestyle, a validation protocol for fitness trackers is needed [21].

Therefore, the aim of this study was to analyse the validity and accuracy of the low-cost MB4 fitness tracker in relation to SC and HR in free-living conditions. Our initial hypothesis is that MB4 offers high values of validity and accuracy with respect to SC and HR.

## 2. Materials and Methods

### 2.1. Participants

In the current study, in total, 46 participants took part in this study. Here, 27 men (mean age: 38.63 ± 11.45 years; height: 175.00 ± 6.38 cm; weight: 70.53 ± 6.70 kg) and 19 women (mean age: 38.68 ± 18.04 years; height: 160.06 ± 6.26 cm; weight: 56.22 ± 9.09 kg) The inclusion criteria were that the individuals: a) did not have any physical limitation that prevented them from performing the validation protocol; b) did not have any treatment that could disrupt the cognitive and physical functions needed to understand the instruction.

Before starting the protocol, all participants gave written informed consent to joining this study. We followed the ethical recommendations approved by the Declaration of Helsinki (2013). Moreover, we followed the directives of the European Union on Good Clinical Practice (111/3976/88 of July 1990), as specified in a national legal framework for human clinical research (Royal Decree 561/1993 on clinical essays). The study was approved by the Ethics Committee of University of Jaén (Protocol code: OCT.20/7.PRY).

### 2.2. Materials

In order to assess SC and HR in laboratory and free-living conditions we used different devices. To compare SC in the laboratory, a video camera was used. To assess the total SC among devices, we compared the MB4 with the Sense Wear Armband (SW). To analyse and compare HR, we used the Firstbeat Body Guard 2 (FB).

On the other hand, the camera used was the Exilim EX-10 (Casio Computer Co., Ltd., Tokyo, Japan), which has a number of effective pixels of 12.1 megapixels (/million), and an image sensor of 1/1.7-inch high-speed CMOS (back-illuminated type). A Full HD resolution 1920 × 1080 (30 fps) was used. Moreover, the maximum digital zoom (31.2X) and auto focus options were chosen to record the total track performed by the participants and step number in as high a quality as possible.

The SW (BodyMedia Inc., Pittsburgh, PA, USA) is a multisensory portable device that provides information about energy expenditure, sleep, circadian rhythms, SC, and activity intensity. This device has been used in previous studies [22,23]. The SW was located on the non-dominant arm between the acromion and olecranon, in accordance with the manufacturer’s recommendation. SenseWear^TM^ software (Pro version 8.1, BodyMedia, Pittsburgh, PA, USA) was used to analyse the data.

The FB (Firstbeat Technologies Ltd., Jyväskylä, Finland) is a beat-to-beat HR monitoring device that is designed for long-term monitoring of HRV and PA. Thanks to the electrodes, FB records the electrocardiogram (ECG), processes the signal with an integrated algorithm, and provides beat-to-beat R-to-R interval (RRI) as an output with 1 ms resolution. The FB has been compared with standard clinical ECG-derived RRI during rest and different conditions of PA for proving its accuracy. Moreover, it detects on average 99.95% of the heartbeats compared to the ECG (gold standard) [24]. Data extraction was made through the Uploaded desktop tool for Windows (Firstbeat) and data analysis was performed through Kubios HRV standard software for Windows (version 3.5). This software is free of charge and it can download from the official website: http://kubios.uef.fi, accessed on 22 December 2021, [25].

The MB4 (Xiaomi Corp., Beijing, China) is a wristband activity tracker considered as a low-cost wearable (~25 EUR). It was placed on the non-dominant wrist following the manufacturer’s indications. It quantifies the number of steps through 3-D accelerometers, as well as a 3-D gyroscope, whereas HR is detected with a photoelectric sensor. The MB4 was connected via Bluetooth to the mobile app Mi Fit (Huami Co., Ltd., Hefei, China).

### 2.3. Procedures

The MB4 validation was made in adults in laboratory and free-living conditions. In laboratory conditions. The SC recorder using MB4 were compared with video recording both at low and high gait speed. Nevertheless, in free-living conditions SC were compared with the SW, whereas HR was compared to the FB.

#### 2.3.1. Laboratory Conditions Protocol

**Video protocol.** To compare the step number registered by MB4 and those performed by the participants, a specific track was designed. We used a high-resolution video camera (Casio Exilim EX-10, Casio, Tokyo, Japan) to record the total steps performed by the participants. The video camera was positioned at an adequate distance for recording the track (i.e., 38 m) (Figure 1). The track consists of a distance of 50 m (straight line) which was delimitated by cones (start and endpoint). Participants started from the initial point (A) to the midpoint (B) (50 m) and turned around until they reached the initial point again (100 m).

The participants had to complete the track two different times, as well as at varying paces. The instruction for the first time was: “you have to complete the track without modifying your usual walking pace”. Subsequently, they completed the same distance, but this time they had to walk faster than the first time. The participants could auto-select the pace at which to complete the track and the instructions to complete it were: “you should walk as quickly as possible, but without starting to run”. The MB4 was positioned on the non-dominant wrist and the number of steps registered by the MB4 was noted at the beginning and at the end of the track. In both cases, the number of steps tracked by the MB4 was counted (initial steps minus final steps) and was compared to the number of steps that the investigator counted from the video recording. Both results were subjected to statistical analysis.

#### 2.3.2. Free-Living Condition Protocol

During the free-living conditions, three different protocols were conducted to compare SC, HR.

**Gait in different intensity protocols.** This protocol was conducted outside with the purpose of simulating free-living situations where participants have to walk in daily life. Before beginning the test, all protocol details were explained to participants to avoid any questions and/or interruption during the protocol. Once the participant understood all the instructions, they were equipped with the FB, the SW on the non-dominant forearm, and the MB4 on the non-dominant wrist (following the manufacturer’s instructions for each device). Subsequently, the protocol started, recording SC and HR in different situations. Firstly, HR was measured for five minutes in resting conditions (sitting comfortably for five minutes). Secondly, after a standard warning up, participants performed a track which consisted of up 10 steps and down 10 stairs (i.e., the participants start and finish the track in the same point (20 stairs). In this case, 10 repetitions of this track were performed (200 stairs in total), and an auto-selected pace was allowed. The time to complete the track, the number of steps, and the HR were recorded. Thirdly, the SC at different paces was recorded. This test was conducted on a straight and flat surface. A metronome app was installed on the participant’s mobile phone that indicated different paces sequences. The pace was maintained for five minutes and the step frequencies were 70–90–110–130 steps per minute [3]. The investigator accompanied the participants to check that their pace was in concordance with the metronome’s pace (Figure 2).

**24-h SC protocol.** Participants were asked to wear the SW on the non-dominant forearm (only removing it for showering or swimming) and the MB4 on the non-dominant wrist for 24 h uninterruptedly to record the SC for both devices. After 24 h, participants were asked to remove the SW and MB4 for the data recording to be analysed [21].

### 2.4. Statistical Analysis

The data were analysed using SPSS, v.19.0 for Windows (SPSS Inc., Chicago, IL, USA). The significance level was set at *p* < 0.05. Descriptive data are reported in terms of means and standard deviations (SD). Tests of normal distribution and homogeneity (Kolmogorov-Smirnov and Levene’s tests) were conducted on all data before analysis. Differences between devices were analysed using the student’s t-test. Additionally, the effect sizes for group differences were expressed as Cohen’s d; the effect sizes were reported as: trivial (<0.2), small (0.2–0.49), medium (0.5–0.79), and large (≥0.8) [26]. The reliability of the tests was analysed using intraclass correlation coefficients (ICC) and the Bland–Altman graphic, a method for quantifying agreement between two quantitative measurements by constructing limits of agreement. These statistical limits are calculated by using the mean and the standard deviation of the differences between two measurements. Convergent validity was investigated with a Pearson correlation between devices.

## 3. Results

Table 1 and Table 2 shows the validation statistics for the number of steps. The MB4 wristband shows high value of convergent validity with the SW monitor during the 24 h recording. When compared to the baseline value of step frequency, the MB4 wristband shows better accuracy, especially at low and medium speeds, than the SW monitor.

In addition, Figure 3 shows the scatter plots of the steps recorded by the MB4 wristband and the SW over 24 h (left) and at 90 steps per minute (right), giving R^2^ = 0.669 and R^2^ = 0.354, respectively.

Figure 4 shows the number of steps obtained in both comfortable and fast walking by the MB4 and the target count by video, with no significant differences in any condition. We also obtained a Pearson correlation coefficient between the SC of MB4 and SC by video in comfortable walking of r = 0.665, *p* < 0.001 and in fast walking of r = 0.759, *p* < 0.001, as well as an ICC of 0.791 (0.619–0.885), *p* < 0.001 and 0.862 (0.749–0.924), *p* < 0.001, respectively.

Bland-Altman plots in the process of calculating the concordance between MB4 and SW in the different protocols are showed in Figure 5.

On the other hand, Table 3, Figure 6 (Scatter plots), and Figure 7 (Bland–Altman plots) show the validation statistics for HR in the different protocols designed. The MB4 wristband showed high values of convergent validity in relation with the FB monitor during recording at rest and in all walking situations.

## 4. Discussion

The aim of this study was to analyse the validity and accuracy of the low-cost MB4 fitness tracker in relation to SC and HR in free-living conditions. Our results confirm our initial hypothesis, since the MB4 is a valid and feasible low-cost fitness tracker in relation to SC and HR. In addition, we have developed a novel validation protocol that could be used in future validation studies of these devices.

Regarding the SC, the current study found that the MB4 is an accurate device for quantifying steps specially in laboratory setting in an adult population. The results that we obtained match those with observed in earlier studies which obtained similar values in previous versions of this wearable (i.e., MB3, and MB2) in relation to SC [18,19,21,27,28].

Concerning laboratory conditions, high accuracy was found in relation to the number of steps between MB4 and the number of steps registered through video recorder at both a comfortable and a high-velocity gait. These results differ from previous studies since the authors concluded that the MB2 underestimated steps at slower walking speeds (<2.88 km/h) [27]. Similar results were also found during the validation of the MB3, since Topalidis et al. concluded that the MB3 underestimated the number of steps, especially in a day with few steps [29].

Regarding the free-living condition protocol, MB4 did not show a proper precision. However, there is not a gold standard protocol to compared MB4 with another device based on accelerometry to everyday use. Nevertheless, the MB4 showed that it has a higher accuracy than the SW when the SC is quantified at a lower (60–79 steps/min), and medium walking speed (80–99 steps/min) [3]. To our knowledge there is previous research that validates the MB4 SC, but it was conducted only in older adults [28]. Consequently, we cannot compare our results with the previous one. Nevertheless, our results match with those of the previous versions of the MB (i.e., the MB2, and MB3) since the authors concluded that the MB can be considered as a suitable low-cost tracker for measuring SC in free-living conditions. It should be noted that the MB2 underestimated steps at slower walking speeds (<2.88 km/h) [27], and the MB3 underestimated the step number especially in days with few steps [29]. On the contrary, Pino-Ortega et al. [28] found that the SC in the MB4 had a high accuracy when it was compared with a GPS, whereas the total distance measured by the MB4 can be considered questionable. A reason for these differences among studies can be explained by that fact that the MB4 does not allow for adjusting the stride length. The MB4 considers this parameter from the height of the athlete, underestimating the total distance.

In terms of HR, the MB4 obtained a high convergent validity when it was compared with the FB during rest time, as well as in the different gait conditions assessed [i.e., slow walking (60–79 steps/min), medium walking (80–99 steps/min)]. This also accords with earlier studies which showed that the preceding version of the MB (i.e., MB2 and MB3) had a high HR accuracy during rest time [27,30,31]. On the contrary, previous versions of these wearables showed an underestimation of HR during a running treadmill, jogging, and cycle ergometer protocol [18,27,31]. These differences between rest time and exercise can be related to the use of photoplethysmography and the algorithm that the MB4 uses to assess it. These problems (HR recording) have been reported previously in different studies with previous versions of the MB, as well as with other low-cost wearables [27,32]. Nevertheless, Chow and Yang [31] concluded that these wearables can be used during daily activities by people at any age with relative accuracy and feasibility.

### 4.1. Limitations and Strength

A strength of this study is a new protocol that mixes laboratory and free-living conditions to analyse the accuracy of different wearables. Furthermore, our protocol is easy to conduct and can be performed by people of any age. However, there are some limitations in our study that need to be mentioned. Firstly, we did not use a GPS to measure a more accurate SC, especially during the free-living protocol, since it cannot work properly inside a laboratory. Secondly, although we assessed SC in different intensities, we did not measure the number of steps in clinical populations (e.g., people with obesity or disabilities) who can have gait alterations depending on the pathology. Thirdly, in outdoor environments and free-living conditions, it is difficult to find a gold standard protocol to compare the MB4, although the lab tests used in the current study, have made up for this limitation. Finally, the evaluation of the accuracy of MB4 has only been carried out in several tests at low speed and low activity, and in a minimum time on the tracker.

### 4.2. Practical Applications

This protocol is original and novel. In addition, it is organized on the basis of the most common motor events in older people.

## 5. Conclusions

To conclude, the validity and precision of the MB4 is reasonable and can be used to monitor the average of SC and HR in free-living conditions, and an indoor and outdoor environment. Taken together, our findings confirm that MB4 is a useful device to measure SC and HR daily. However, the current study did not evaluate reliability, test economics, nor ease of use.

## Figures and Tables

**Figure 1 ijerph-19-01593-f001:**
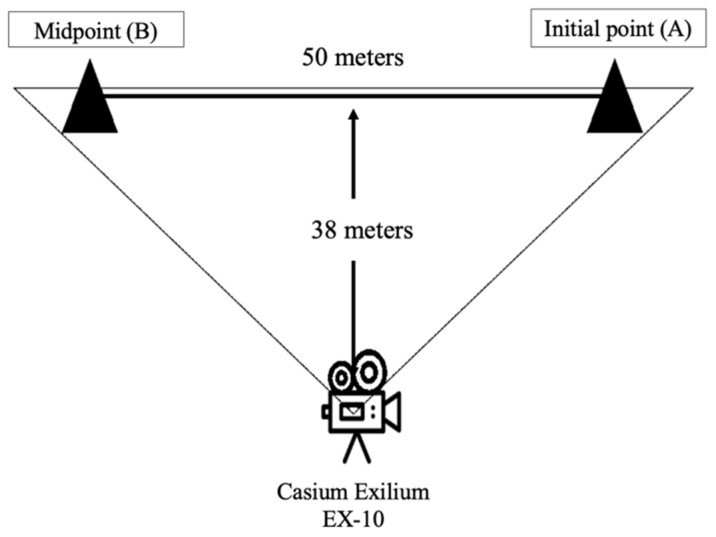
Laboratory condition protocol.

**Figure 2 ijerph-19-01593-f002:**
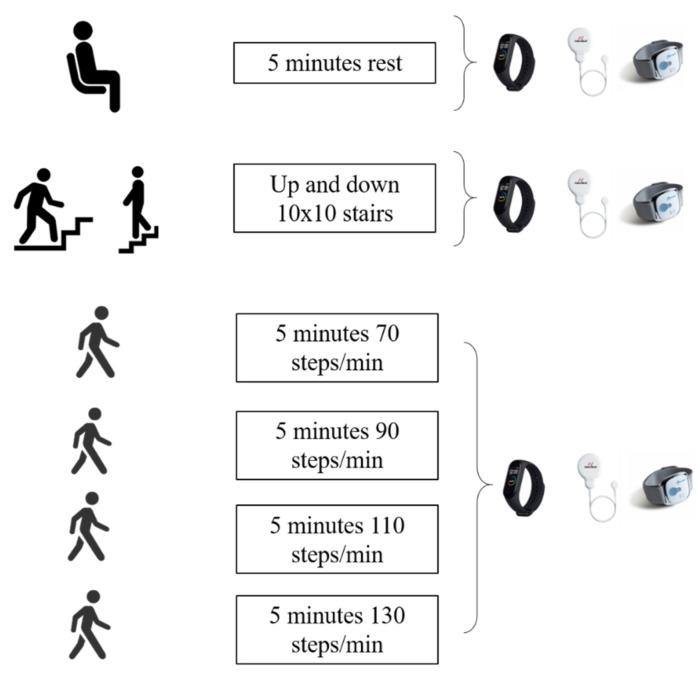
MB4 validation protocol in free-living condition.

**Figure 3 ijerph-19-01593-f003:**
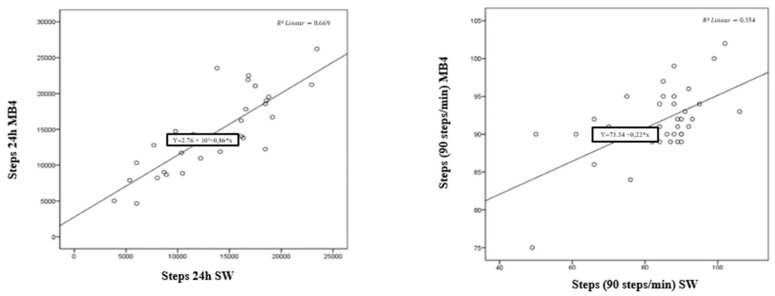
Scatter plots of the steps recorded by the MB4 and SW wristband over 24 h (**left**) and at 90 steps per minute (**right**).

**Figure 4 ijerph-19-01593-f004:**
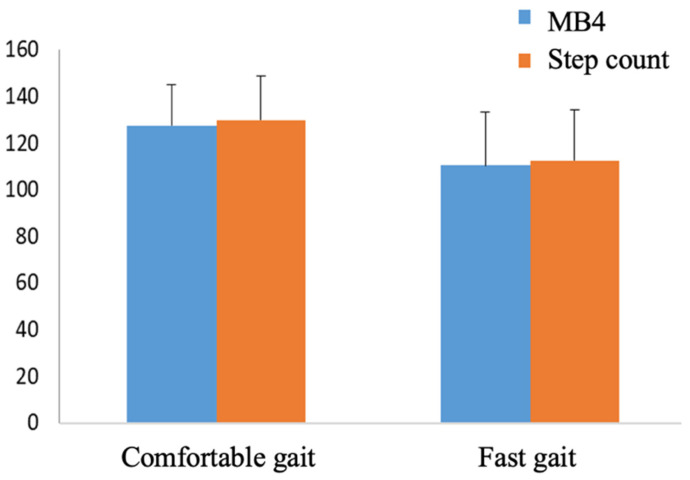
Number of steps obtained in comfortable and fast gait by MB4 and the target count by video.

**Figure 5 ijerph-19-01593-f005:**
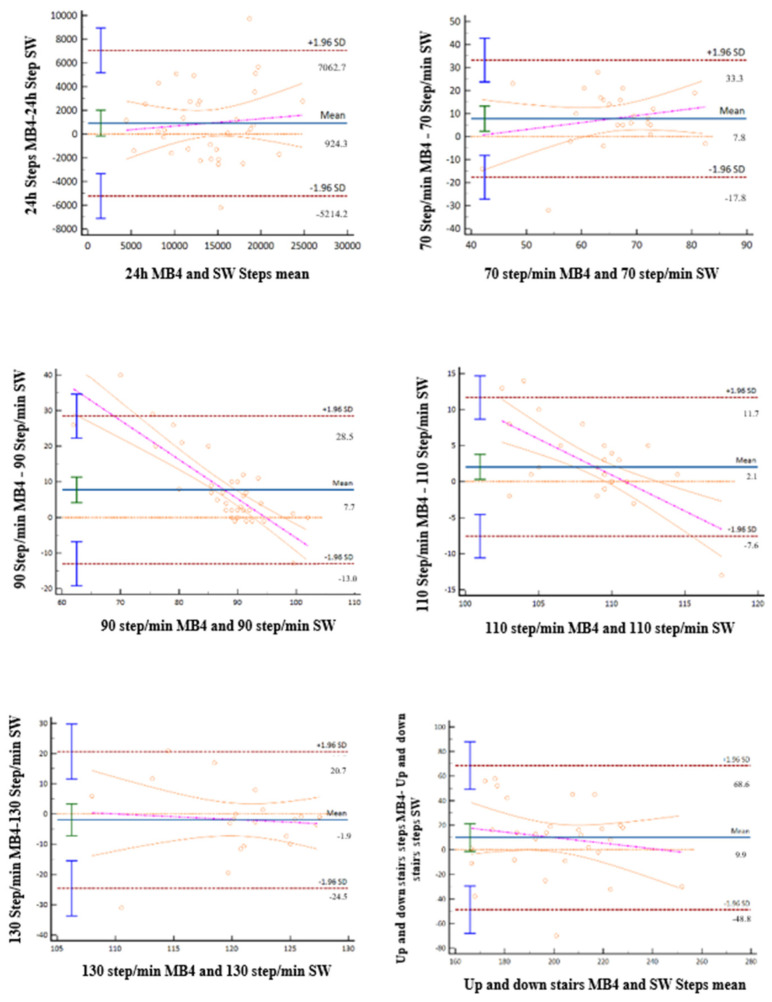
Bland-Altman plots in the process of calculating the concordance between MB4 and SW in the different gait protocols.

**Figure 6 ijerph-19-01593-f006:**
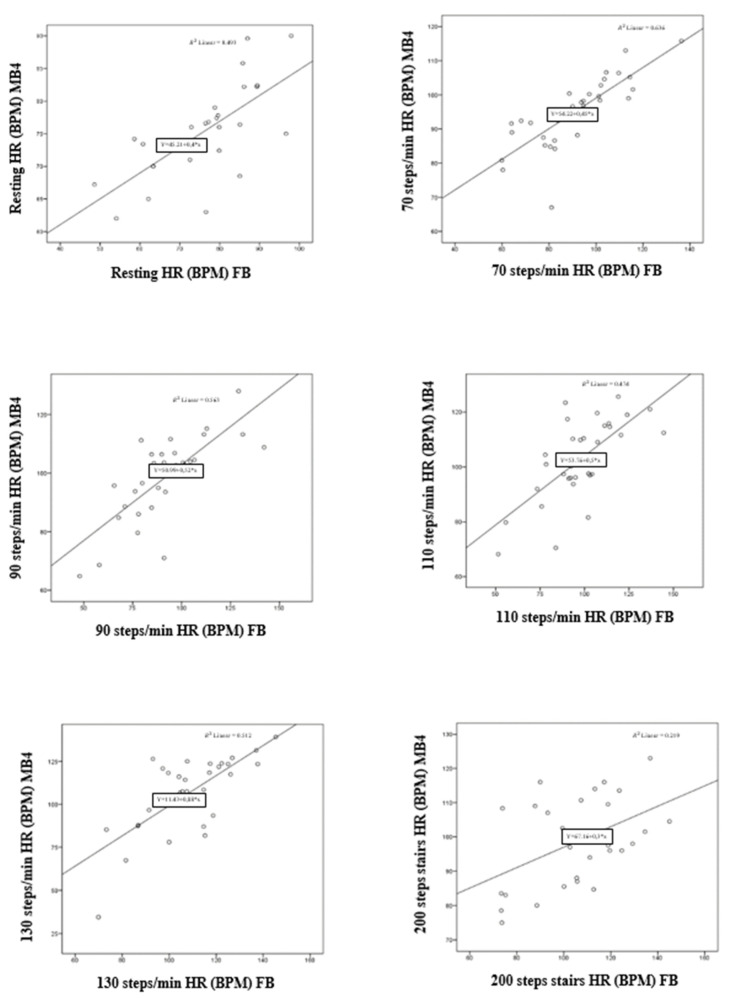
Scatter plots of HR recorded by the MB4 wristband and FB heart sensor during (from top to bottom and from left to right): resting, walking at 70–90–110–130 steps per minute and walking up and down 200 stairs.

**Figure 7 ijerph-19-01593-f007:**
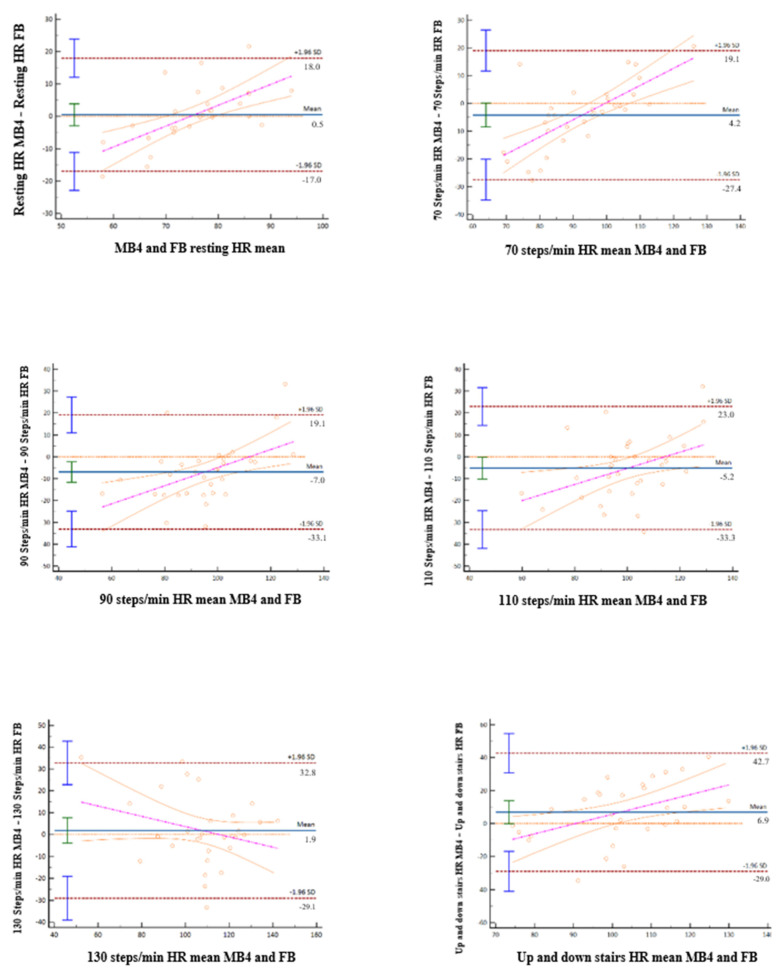
Bland-Altman plots in the process of calculating the concordance between MB4 and FB at rest and in the different gait protocols.

**Table 1 ijerph-19-01593-t001:** Validation statistics of the steps recorded by comparing the SW monitors and the Xiaomi MB4 wristband.

Difference Recording Tasks	SW Mean (SD)	MB4 Mean (SD)	*p*-Value	Cohen’s -d	Pearson Correlation Coefficient	ICC	*p*-Value
Number of steps	13,536.11 (5020.98)	14,460.38 (5309.96)	0.095	0.178	0.818 ***	0.899 (0.798–0.950)	<0.001
70 steps per minute	61.62 (10.04)	69.41 (12.33)	0.007	0.692	0.340	0.500 (−0.157–0.784)	0.052
90 steps per minute	83.88 (12.67)	91.63 (4.64)	<0.001	0.812	0.595 ***	0.556 (0.128–0.773)	0.009
110 steps per minute	107.84 (5.19)	109.90 (2.45)	0.022	0.507	0.349 *	0.425 (−0.165–0.716)	0.061
130 steps per minute	121.73 (8.25)	119.81 (7.59)	0.454	0.242	−0.057	−0.121 (−1.762–0.545)	0.599
200 steps in stairs	194.73 (28.25)	204.63 (24.19)	0.081	0.376	0.356 *	0.521 (−0.007–0.772)	0.026

70, 90, 110, and 130 steps: cadence of the steps per minute at different intensities; 200 steps: number of steps measured by SW and MB4 during stairs test (upstairs and downstair); SD: standard deviation; * *p* < 0.05; *** *p* < 0.001.

**Table 2 ijerph-19-01593-t002:** Adjustment statistics of the number of steps by the T-test for a sample in relation to the reference steps.

Step Reference	SW Mean Difference/*p*-Value	MB4 Mean Difference/*p*-Value
200 steps in stairs	−6.78/0.196	4.63/0.303
70 steps per minute	−8.37/<0.001	−0.58/0.819
90 steps per minute	−6.11/0.007	1.63/0.042
110 steps per minute	−2.15/0.024	−0.11/0.779
130 steps per minute	−8.26/<0.001	−10.18/<0.001

SW: Sense Wear; MB4: Mi band 4.

**Table 3 ijerph-19-01593-t003:** Validation statistics of HR recorded by comparing FB monitors and Xiaomi MB4 wristband.

Heart Rate at Different Intensities	FB Mean (SD)	MB4 Mean (SD)	*p*-Value	Cohen’s -d	Pearson Correlation Coefficient	CCI (Confidence Interval)	*p*-Value
HR (Bpm) at rest	75.68 (12.31)	75.18 (6.94)	0.768	0.050	0.702 ***	0.751 (0.469–0.883)	<0.001
HR (Bpm) at 200 stairs	105.48 (20.30)	98.62 (12.94)	0.053	0.402	0.468 **	0.596 (0.139–0.810)	0.010
HR (Bpm) at 70 steps/min	90.68 (18.31)	94.87 (10.29)	0.054	0.282	0.797 ***	0.810 (0.612–0.907)	<0.001
HR (Bpm) at 90 steps/min	91.62 (20.05)	98.58 (13.38)	0.005	0.408	0.750 ***	0.825 (0.646–0.914)	<0.001
HR (Bpm) at 110 steps/min	97.56 (19.75)	102.73 (14.40)	0.043	0.299	0.688 ***	0.792 (0.583–0.896)	<0.001
HR (Bpm) at 130 steps/min	108.30 (18.23)	106.42 (22.35)	0.513	0.092	0.715 ***	0.824 (0.635–0.915)	<0.001

FB: First Beat; MB4: Mi band 4; HR: Heart Rate; SD (standard deviation); Bpm: Beats per minute; ** *p* < 0.01; *** *p* < 0.001.

## Data Availability

Not applicable.

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
