# Peer review of "Is the Xiaomi Mi Band 4 an Accuracy Tool for Measuring Health-Related Parameters in Adults and Older People? An Original Validation Study"

_ijerph, 2022, doi:10.3390/ijerph19031593_

Round 1

Reviewer 1 Report

This paper presents a validation study of the Mi Band 4 in terms of measuring steps counts, heart rate and total sleep time. I like the idea of performing validation in both laboratory settings and naturalistic settings. Other than that, I don't see much originality except that the cohort seems to be younger than prior studies. I understand that it's difficult to implement a lab-based gold standard as the ground truth (e.g., the video based approach for step counting, or PSG for sleep monitoring) in naturalistic settings, and it's plausible to use alternative medical-grade devices. However, it's not straightforward which device was used as the ground truth to compare the Mi Band with for each of the metrics measured. The writing needs significant editing. If the SW was used as the reference device in the validation of step count in a naturalistic setting, then it makes no sense to state that 'MB4 wristband shows better accuracy than the SW monitor'. This statement sounds like there's a third device that was used as the ground truth, which didn't exist. 

The validation of the total sleep time is extremely problematic. I have no idea why it's necessary to compare the SW device with the ActiGraph in terms of sleep time. The focus of interest is to assess the MB4, not the SW. I'd think the SW should not even be involved in sleep assessment, because your ground truth here is the ActiGraph and the only thing need is to compare the MB4 with the ActiGraph. Figure 8 really doesn't tell much about the accuracy of the MB4 on TST. I'd like to see at least a Bland-Altman plot. The assessment on TST feels like a piece of chicken ribs. Probably eliminating the assessment on TST could somehow strengthen the focus of the paper, as it requires more data analysis to thoroughly evaluate the accuracy of MB4 on sleep measurement, and TST is really only one of the important sleep metrics (in practice, researchers would care about the device's accuracy in measuring WASO, SE, the ratio of each sleep stage in addition to TST). I strongly suggest referring to [1] for a better and standard validation protocol for sleep metrics. 

[1] Luca Menghini, Nicola Cellini, Aimee Goldstone, Fiona C Baker, Massimiliano de Zambotti, A standardized framework for testing the performance of sleep-tracking technology: step-by-step guidelines and open-source code, Sleep, Volume 44, Issue 2, February 2021, zsaa170, https://doi.org/10.1093/sleep/zsaa170.

Author Response

REPLY TO REVIEWER

We appreciate very much your constructive comments, helpful information and your time. Thanks to this review, our manuscript was substantially improved. Responses to your comments are written in bold.

Reviewer 1

This paper presents a validation study of the Mi Band 4 in terms of measuring steps counts, heart rate and total sleep time. I like the idea of performing validation in both laboratory settings and naturalistic settings. Other than that, I don't see much originality except that the cohort seems to be younger than prior studies. I understand that it's difficult to implement a lab-based gold standard as the ground truth (e.g., the video based approach for step counting, or PSG for sleep monitoring) in naturalistic settings, and it's plausible to use alternative medical-grade devices. However, it's not straightforward which device was used as the ground truth to compare the Mi Band with for each of the metrics measured. The writing needs significant editing. If the SW was used as the reference device in the validation of step count in a naturalistic setting, then it makes no sense to state that 'MB4 wristband shows better accuracy than the SW monitor'. This statement sounds like there's a third device that was used as the ground truth, which didn't exist. 

Thank you for your suggestions. You are right to indicate that the devices used were not a good gold standard, however in outdoor environments and free-living conditions, it is difficult to find a gold standard to compare the MB 4, although the lab tests used in the current study, have made up for this limitation. In particular, the differences between MB4 and SW could be due to the different location of each device, on the wrist and on the forearm, respectively. Although, the Bland-Altman analysis indicates a good agreement between these two devices.

The validation of the total sleep time is extremely problematic. I have no idea why it's necessary to compare the SW device with the ActiGraph in terms of sleep time. The focus of interest is to assess the MB4, not the SW. I'd think the SW should not even be involved in sleep assessment, because your ground truth here is the ActiGraph and the only thing need is to compare the MB4 with the ActiGraph. Figure 8 really doesn't tell much about the accuracy of the MB4 on TST. I'd like to see at least a Bland-Altman plot. The assessment on TST feels like a piece of chicken ribs. Probably eliminating the assessment on TST could somehow strengthen the focus of the paper, as it requires more data analysis to thoroughly evaluate the accuracy of MB4 on sleep measurement, and TST is really only one of the important sleep metrics (in practice, researchers would care about the device's accuracy in measuring WASO, SE, the ratio of each sleep stage in addition to TST). I strongly suggest referring to [1] for a better and standard validation protocol for sleep metrics. 

You are right, sleep analysis has been removed.

[1] Luca Menghini, Nicola Cellini, Aimee Goldstone, Fiona C Baker, Massimiliano de Zambotti, A standardized framework for testing the performance of sleep-tracking technology: step-by-step guidelines and open-source code, Sleep, Volume 44, Issue 2, February 2021, zsaa170, https://doi.org/10.1093/sleep/zsaa170.

Reviewer 2 Report

Brief summary

This paper investigates the validity and reliability of the Xiomi Mi Band 4 fitness tracker’s ability to characterize total step count, total sleep time and heart rate.  The paper cites a secondary aim of validating a protocol to assess this validity and reliability.  The paper uses a novel testing technique, using both an indoor/laboratory condition as well as “free living conditions” out of the laboratory. 46 healthy adults were assessed in these conditions while simultaneously wearing the MI 4 band, as well as an Actigraph Link step counter, a Firstbeat Body Guard 2 heart rate evaluation tool and a Sense Wear Arm band to assess total sleep time. Statistical analysis of the similarities in data output from these devices was evaluated.

Broad Comments

  • The paper is well written, understandable, and well-organized.
  • This reliability study, though, is missing an assessment of reliability. No test-retest measurements were assessed.
  • The validity of the fitness band was assessed compared to a two “other low cost” fitness trackers.Correlations in heart rate measurements were supported, but I am not sure the power of the study was strong enough to support the conclusions asserted for step counts. ICC and Pearson correlation coefficients were only “fair” agreement, likely due to the small numbers of steps and trials.
  • Total sleep time was significantly different between each tracking device and no significant findings could be made. The paper appropriately cites the inability to use laboratory polysomnography device, but only concludes what was cited in other studies.
  • While the paper presents a secondary aim of validating a protocol to assess validity and reliability, this protocol is not validated in any way in this study. The study does not validate this protocol versus other protocols.  Publishing this study will establish this as a different protocol, but this protocol still needs to stand up to further peer review over time. This secondary aim should be removed.  I would like to see a greater evaluation of the strengths and limitations of this new protocol in the discussion.

Specific comments

  • Lines 16 I am not sure to what “concurrent” validity and reliability refers.    
  • Line 18 Secondary aim should probably be removed and discussed further in the discussion.
  • Line 27-28 Reconsider what are appropriate conclusions.  This validity test did not test “usefulness”, or “reliability”.  “Ease of use” was also not evaluated “for people of any age” as only healthy adults were evaluated.  I am not sure there is enough power with the minimum steps counted in the laboratory to make any meaningful concusions about validity
  • Lines 52 Is “induce” the correct word
  • Line 70 various instead of each
  • Line 98 nineteen
  • Line 128 What is Professional 8.1 software? (company, location)
  • Line 168-170 Is walking in a straight line for a distance of 100 m a reasonable test to assert validity between devices?  Most other studies use at least 30 minutes of activity.
  • Line 215 Cite specific studies
  • Table 1 Make sure to explain the low ICC values and adjust conclusion appropriately.
  • Table 1 and Table 4 I don’t understand what the P values attached to the Pearson Correlation Coefficient mean (indicated by the asterisks).  What is this p-value comparing and how was it calculated?  Similarly, what is the p-value on the far right?
  • Line 289 Validating protocol not accomplished
  • Line 291 Define valid. 
  • Line 292 Discuss how these trackers were selected. Why not use a hip based step count tracker that has been found to be more accurate than arm based counters?  Why only two trackers and not more? 
  • Line 293,298 Has “accuracy” been measured in this study?
  • Line 335 Re-evaluate whether TST should be included at all in this study.
  • Line 349 Limitations need to be expanded in many ways: reliability, only low speed and low activity testing, data loss, type of trackers used, no gold standard, protocol uses minimal time in the tracker, lack of power evaluation
  • Line 365 re-evaluate conclusion

Author Response

We appreciate very much your constructive comments, helpful information and your time. Thanks to this review, our manuscript was substantially improved. Responses to your comments are written in bold.

Reviewer 2

Broad Comments

  • The paper is well written, understandable, and well-organized.
  • This reliability study, though, is missing an assessment of reliability. No test-retest measurements were assessed.
  • The validity of the fitness band was assessed compared to a two “other low cost” fitness trackers.Correlations in heart rate measurements were supported, but I am not sure the power of the study was strong enough to support the conclusions asserted for step counts. ICC and Pearson correlation coefficients were only “fair” agreement, likely due to the small numbers of steps and trials.
  • Total sleep time was significantly different between each tracking device and no significant findings could be made. The paper appropriately cites the inability to use laboratory polysomnography device, but only concludes what was cited in other studies.
  • While the paper presents a secondary aim of validating a protocol to assess validity and reliability, this protocol is not validated in any way in this study. The study does not validate this protocol versus other protocols.  Publishing this study will establish this as a different protocol, but this protocol still needs to stand up to further peer review over time. This secondary aim should be removed.  I would like to see a greater evaluation of the strengths and limitations of this new protocol in the discussion.

Thank you for your suggestions. You are right to indicate that reliability was not evaluated of MB4. In this regard, we have rewritten the purpose of this study because we have really analysed the validity and precision of this device. In addition, we have removed the second purpose of our study, which is related to protocol validation. Also, although the concurrent validation with other devices was not strong, the comparison with video recording under laboratory conditions was good. Certainly, the analysis of TST was not good, however, this is a finding. Finally, other limitations were added.

Specific comments

  • Lines 16 I am not sure to what “concurrent” validity and reliability refers.    

Thank you for your comment. The purpose is confusing; we have rewritten it.

The aim of this study was to analyse the validity and accuracy of the low-cost Xiaomi Mi Band 4 (MB4) fitness tracker in relation to step count and heart rate in free-living conditions.

Line 18 Secondary aim should probably be removed and discussed further in the discussion.

Done

  • Line 27-28 Reconsider what are appropriate conclusions.  This validity test did not test “usefulness”, or “reliability”.  “Ease of use” was also not evaluated “for people of any age” as only healthy adults were evaluated.  I am not sure there is enough power with the minimum steps counted in the laboratory to make any meaningful concusions about validity

You are right conclusions section has been rewritten.

 Regarding the use of MB4 only in healthy adults, noticed that we mention this limitation in our paper (line 356) since the protocol have not been conducted in clinical population.

In relation to the number of steps during the laboratory protocol condition, we have considered enough distance to conduct our protocol based on previous research. See:

Beltrán-Carrillo, V. J., Jiménez-Loaisa, A., Alarcón-López, M., & Elvira, J. L. (2019). Validity of the “Samsung Health” application to measure steps: A study with two different samsung smartphones. Journal of sports sciences, 37(7), 788-794.

  • Lines 52 Is “induce” the correct word

We appreciate your comment. We have re-written the sentence and we hope that now is clearer than before. “In this sense, a walking programme based on the number of steps assessed by a pedometer can motivate to increase PA level in a long-term”…

  • Line 70 various instead of each

Done

  • Line 98 nineteen

We apologise for this mistake. We have change it.

  • Line 128 What is Professional 8.1 software? (company, location)

Thank you for your comment. We are talking about the version of SenseWear software that we used to analyse the data. Anyway, we have added more information to clarify the sentence.

…SenseWear™ software (Pro version 8.1, BodyMedia, PA) was used to analyse the data.

Line 168-170 Is walking in a straight line for a distance of 100 m a reasonable test to assert validity between devices?  Most other studies use at least 30 minutes of activity.

Thank you for your comment. We are not really sure about which studies are you referring; It should be notice that it is in a laboratory condition, and similar distance (even less) have been used before to conduct an android’s app validation protocol (Beltran-Carrillo et al. 2018)

Hence, we consider that 100 m is a reasonable distance since the MB4 start to count at the same moment that the participants start to walk in laboratory setting. Thus, it is enough distance to measure and compare the number of steps between MB4 and video recorder. In addition, the same protocol was conducted two times at different intensities, it allows us to compare the numbers of steps in different gait conditions and get a more accuracy information (Beltran-Carrillo et al. 2018).

  • Line 215 Cite specific studies

We appreciate your comment. We have added it.

Reference [21]

Degroote, L.; Hamerlinck, G.; Poels, K.; Maher, C.; Crombez, G.; Bourdeaudhuij, I. De; Vandendriessche, A.; Curtis, R.G.; DeSmet, A. Low-Cost consumer-based trackers to measure physical activity and sleep duration among adults in free-living conditions: Validation study. JMIR mHealth uHealth 2020, 8, 1–22, doi:10.2196/16674.

  • Table 1 Make sure to explain the low ICC values and adjust conclusion appropriately.

You are right, we have rewritten the explanation of the results and the conclusions as:

The MB4 wristband shows high and moderate values of convergent validity with the SW monitor during the 24 h recording, at medium speeds, and climbing stairs respectively.

Table 1 and Table 4 I don’t understand what the P values attached to the Pearson Correlation Coefficient mean (indicated by the asterisks).  What is this p-value comparing and how was it calculated?  Similarly, what is the p-value on the far right?

P values attached to the Pearson Correlation Coefficient indicated the signification levels of these analysis, likewise, the p-value on the far right is the signification level related to the F test in the intraclass correlation analysis. These data were obtained using SPSS software.

  • Line 289 Validating protocol not accomplished

You are right, this line has been rewritten as:

The aim of this study was to analyses the validity and accuracy of the low-cost Xiaomi Mi Band 4 (MB4) fitness tracker in relation to SC and HR in free-living condi-tions.  Our results confirm our initial hypothesis, since the MB4 is a valid and feasible low-cost fitness tracker in relation to SC, HR.

  • Line 291 Define valid. 

We have considered the term valid as useful or accurate.

  • Line 292 Discuss how these trackers were selected. Why not use a hip based step count tracker that has been found to be more accurate than arm based counters?  Why only two trackers and not more? 

We have used these devices since they were the ones that were available in our laboratory.

  • Line 293,298 Has “accuracy” been measured in this study?

Perhaps we have not expressed the objective of this study so well, we have really analyzed the accuracy and validity of Mi Band 4 with respect to other devices and real data.

  • Line 335 Re-evaluate whether TST should be included at all in this study.

Thank for your suggestion, sleep analysis has been removed.

  • Line 349 Limitations need to be expanded in many ways: reliability, only low speed and low activity testing, data loss, type of trackers used, no gold standard, protocol uses minimal time in the tracker, lack of power evaluation

Thank for your suggestion, we have added some of these limitations.

  • Line 365 re-evaluate conclusion

Done

Round 2

Reviewer 1 Report

I appreciate the efforts that the authors put into addressing my comments. The paper has been improved substantially. 

Author Response

Thank you for your comment. We have checked the English’ grammar through the manuscript to improve our paper. It should be noted that our manuscript has been rewiever by a Proof-Reading-Service to avoid misunderstanding information related with research. You can find a certificate attached.

Please, if you consider that there is a specific word/sentence that you consider that have to be change, you feel free to tell us.

Thank you so much in advance.

Reviewer 2 Report

The authors have made significant improvements to the readability and accuracy of the paper and in many ways have addressed my previous comments. In particular, the presentation of heart rate data is much more understandable.  The conclusions from this section of the paper seem appropriate and valid.

I still have concerns, though, about the paper's conclusions with regards to step counts.  With the exception of the 24 hour step count, correlations between the two fitness counters are relatively poor (PCC < .6 - and even negative in some cases) (ICC <.6 with error ranges very low and even negative in most cases).  It is hard to understand single line conclusions citing 'high and moderate values of validity" given these findings.  It seems the testing protocol is not rigorous enough to demonstrate better correlation, or that there is too much difference in the recorded steps to make any conclusions.  This at least needs to be discussed in the conclusion, which is currently absent. The conclusion also continues to refer to the accuracy of the MB4 (Line 273) when no gold standard was used to determine which was more accurate.

The final conclusion (Line 316-320) also continues to overstate the findings of this study.  The "accuracy" of these devices were not studied. The conclusions need to restate that the population studied here was younger healthy subjects.  And in the last line, the study did not evaluate reliability, test economics, nor ease of use. 
